# Chemical Composition and Antibacterial Activity of Liquid and Volatile Phase of Essential Oils against Planktonic and Biofilm-Forming Cells of *Pseudomonas aeruginosa*

**DOI:** 10.3390/molecules27134096

**Published:** 2022-06-25

**Authors:** Malwina Brożyna, Justyna Paleczny, Weronika Kozłowska, Daria Ciecholewska-Juśko, Adam Parfieńczyk, Grzegorz Chodaczek, Adam Junka

**Affiliations:** 1Department of Pharmaceutical Microbiology and Parasitology, Medical University of Wroclaw, 50-367 Wroclaw, Poland; justyna.paleczny@student.umw.edu.pl; 2Division of Pharmaceutical Biotechnology, Department of Pharmaceutical Biology and Biotechnology, Medical University of Wroclaw, 50-367 Wroclaw, Poland; weronika.kozlowska@umw.edu.pl; 3Department of Microbiology and Biotechnology, West Pomeranian University of Technology, 70-311 Szczecin, Poland; daria.ciecholewska@zut.edu.pl; 4Faculty of Medicine, Lazarski University, 02-662 Warsaw, Poland; adam.parfienczyk@lazarski.pl; 5Bioimaging Laboratory, Łukasiewicz Research Network—PORT Polish Center for Technology Development, 54-066 Wroclaw, Poland; grzegorz.chodaczek@port.org.pl

**Keywords:** biofilm, *Pseudomonas aeruginosa*, essential oil, EOs in liquid form, volatile fractions, antimicrobial activity

## Abstract

*Pseudomonas aeruginosa* is an opportunistic pathogen causing life-threatening, hard-to-heal infections associated with the presence of a biofilm. Essential oils (EOs) are promising agents to combat pseudomonal infections because of the alleged antimicrobial activity of their volatile fractions and liquid forms. Therefore, the purpose of this paper was to evaluate the antibacterial efficacy of both volatile and liquid phases of seven EOs (thyme, tea tree, basil, rosemary, eucalyptus, menthol mint, lavender) against *P. aeruginosa* biofilm and planktonic cells with the use of a broad spectrum of analytical in vitro methods. According to the study results, the antibacterial activity of EOs in their liquid forms varied from that of the volatile fractions. Overall, liquid and volatile forms of rosemary EO and tea tree EO displayed significant antibiofilm effectiveness. The outcomes indicate that these particular EOs possess the potential to be used in the therapy of *P. aeruginosa* infections.

## 1. Introduction

*Pseudomonas aeruginosa* is a Gram-negative, opportunistic bacterium responsible for a growing number of serious nosocomial infections. As an example, such different populations as patients suffering from chronic wounds, cystic fibrosis, catheter-related infections, or AIDS are at risk of developing a severe pseudomonal infection, accounting for high mortality and morbidity rates [1]. The production of surface factors, flagella, pili, lipopolysaccharide, toxin secretion and biofilm formation are considered primary virulence determinants of *P. aeruginosa* [2]. Biofilms are complex microbial communities in which cells display characteristic spatial localization and phenotypic and biochemical features differentiating them from their free-swimming (planktonic) counterparts. Cells in the biofilm are encased in extracellular polymeric substances (EPS), which serve as a scaffold for the structural integrity of the microbial community and as a barrier protecting cells from detrimental factors. The biofilm may form on living tissue and indwelling medical devices, including catheters, tracheal tubes, implants, etc. Cells in the biofilm are highly tolerant to antimicrobials (antibiotics, especially) and are resistant to host immune defense mechanisms [3]. Therefore, therapeutic options for the treatment of pseudomonal, biofilm-based infections are limited. Thus, numerous attempts have been made to devise novel strategies for combating *Pseudomonas* infections [2]. Among them, the application of various kinds of phytochemical molecules has been reported to be a promising direction with regard to *P. aeruginosa* biofilm eradication and overcoming this bacteria’s resistance to antibiotics [4].

One of the types of such phytochemical molecules is referred to as essential oils (EOs). These mixtures of plants’ secondary metabolites belong mainly to groups of terpenes, sesquiterpenes, and phenylpropanoids. Their broad spectrum of biological activity covers antibacterial, antifungal, anti-inflammatory, antiviral, antioxidant, and anticancer properties. Overall, EO activity depends on the content of the constituents, which is, to a major extent, impacted by such variables as plant origin, cultivation conditions, and extraction techniques [5]. EOs are characterized by high lipophilicity and volatility. Due to the lipophilic nature, they exhibit an unspecific mode of antimicrobial action, which relies on binding to the cell wall or the membrane and damaging its integrity. Concerning this broad mechanism of action and already proven effectiveness against multidrug-resistant microorganisms/biofilms, EOs are suggested to be the alternative approach to combat bacterial infections [6]. However, the topical administration of EOs may cause allergic reactions and skin irritation; thus, undiluted EOs cannot be applied directly to body parts altered by the infection process [7]. From the manufacturing perspective, EOs’ lipophilic properties also restrain the development of physically stable, non-harmful skin formulations [8]. Therefore, volatile fractions of EOs are more and more considered to be suitable for the treatment of specific bacterial infections (e.g., of the respiratory tract, skin, or wounds) [9].

The already performed research demonstrated that the volatile forms of EOs may display a stronger antimicrobial effect than their liquid phases used in direct contact [10]. It is suggested that hydrophobic molecules in the aqueous phase associate and form micelles, which hinder the attachment of EOs to microorganisms. The EOs’ volatile fractions are devoided of this disadvantage [11]. In addition, because EOs’ volatile fractions may be administered locally (not via systemic dosage), the risk of side effects and interactions with other drugs is exiguous [8]. However, the majority of research is still mainly focused on EOs’ antimicrobial activity through direct contact with the oil in its liquid form. Only a limited number of studies on antimicrobial effectiveness of the volatile forms of EOs are available [12]. Therefore, the purpose of this paper was to evaluate the antibacterial potential of both volatile and liquid phases of seven commercially available EOs against *P. aeruginosa* biofilm and planktonic cells with the use of a broad spectrum of analytical in vitro methods. 

## 2. Results

### 2.1. GC–MS (Gas Chromatography–Mass Spectrometry) Analysis of the Tested EOs Composition

The percentage composition of EOs’ constituents was assessed using gas chromatography–mass spectrometry. The two main ingredients of thyme oil (T-EO) are thymol and p-cymene. Terpinen-4-ol and γ-terpinene predominate in tea tree oil (TT-EO). Basil oil (B-EO) is composed of methyl chavicol and linalool only. Camphor and 1,8-cineole are the main components of rosemary oil (R-EO), while 1,8-cineole and γ-terpinene are primarily presented in eucalyptus oil (E-EO). Menthol mint oil (MM-EO) contains mostly menthol and menthone, and linalyl acetate and linalool are the main ingredients of lavender oil (L-EO). The compositions of EOs were compared to the Polish Pharmacopoeia XI standards. TT-EO was the only one whose components (and their percentage) were directly within pharmacopeial ranges. B-EO is not included in the Pharmacopoeia. Table 1 presents a list of the EOs’ detected compounds. Chromatograms of the tested EOs are presented in Appendix A. 

### 2.2. Biofilm Biomass and Activity Level Assessment

In the next line of investigation, the level of *P. aeruginosa* biofilm biomass and its cells’ metabolic activity were assessed. All tested strains were metabolically active and able to form biofilm under applied in vitro conditions; however, differences in both biofilm biomass and metabolic activity were observed between specific strains (Figure 1).

### 2.3. Antimicrobial Activity of EOs in Their Liquid Forms against P. aeruginosa Planktonic Forms

Two standard techniques were performed to evaluate the antimicrobial effectiveness of EOs in their liquid forms toward planktonic cells: disc diffusion, in which non-emulsified EOs were applied, and minimal inhibitory concentration (MIC) estimation, in which EOs emulsions were used. The results from the first methodology mentioned are presented in Table 2. R-EO exhibited the highest antimicrobial activity, and zones of growth inhibition (mm) were obtained for all strains. T-EO and E-EO displayed moderate antibacterial efficacy against most of the strains. L-EO was almost inactive against *P. aeruginosa* strains. Because Tween 20 was used as an emulsifying agent in the MIC assay, its antibacterial potential against planktonic forms of *P. aeruginosa* ATCC 15442 strain was also evaluated. The aforementioned substance did not influence *P. aeruginosa* cell growth in concentrations from 1.0% to 0.002% (*v*/*v*), as it is presented in Appendix A. R-EO emulsion was the most active among the tested EOs against planktonic pseudomonal cells. MIC values of B-EO ranged from 12.5% to 1.6% (*v*/*v*). In other EOs, MIC values were detected mostly in concentrations of 12.5–25.0% (*v*/*v*). Moreover, particular strains were not sensitive to MM-EO and T-EO even if a 25.0% (*v*/*v*) concentration of these oils was applied. The detailed list of the determined MIC (%) values is presented in Table 3.

### 2.4. Antimicrobial Activity of EOs in Their Liquid Forms against P. aeruginosa Biofilms

The antibiofilm activity of EOs in their liquid forms was determined using the standard microdilution method and the modified A.D.A.M. (antibiofilm dressing’s activity measurement) technique. To evaluate the MBEC values (minimal biofilm eradication concentration) of the emulsified EOs in their liquid forms, resazurin (7-hydroxy-3H-phenoxazin-3-one-10-oxide sodium salt) staining was applied as an indicator of the presence of metabolically active cells. Due to the observed inconsistency of the results (data not shown) obtained for strains treated with T-EO and TT-EO, and dyed with resazurin, TTC (2,3,5-triphenyl-tetrazolium chloride) solution was used instead (for these above-mentioned EOs). As the MBEC values were not found within the applied concentration spectrum, changes in the biofilm-forming cells viability (%) after exposure to selected EO concentrations are presented in the study. The significant changes in the biofilm-forming cells viability were assessed after the treatment of pseudomonal biofilm with T-EO and TT-EO emulsions in concentrations of 25.0%–6.3% (*v*/*v*). B-EO and R-EO emulsions, both in concentrations of 25.0%–12.5% (*v*/*v*) and L-EO emulsion in a concentration of 25.0% (*v*/*v*) exhibited potent antibiofilm effectiveness. The percentage changes in the pseudomonal biofilm-forming cells viability after the application of 25.0% (*v*/*v*), 12.5% (*v*/*v*), and 6.3% (*v*/*v*) oils emulsions are presented in Figure 2 and Appendix A, respectively.

To perform the modified A.D.A.M. technique, biocellulose discs were soaked with the non-emulsified EOs. The concentration of EOs released from the biocellulose discs was approximately 65.8%. The biofilms of PA 1–7 and PA 14–16 strains were the most prone to the activity of the EOs in their liquid forms (Table 4). The antibiofilm effectiveness of individual EOs was strain-dependent, although regarding all EOs except for B-EO, the reduction of cells viability equal to 60.11% or more was reported for selected strains. The mentioned B-EO displayed the lowest antibiofilm activity, and 96% (*v*/*v*) ethanol was applied against the reference strain to prove the method’s usability. The mean obtained reduction of the biofilm cells viability after the treatment with ethanol was 90.33%.

### 2.5. Antimicrobial Activity of EOs’ Volatile Fractions against P. aeruginosa

The influence of non-emulsified EOs’ volatile phases on pseudomonal planktonic cells was assessed with the inverted Petri dish method. All of the examined strains (PA 1-7, PA 13–19, ATCC 15442) were resistant to the volatile of L-EO and B-EO. Almost all of the tested strains were resistant to T-EO (PA 1-6, PA 13-19, ATCC 15442), TT-EO (PA 1-7, PA 13–17, PA 19, ATCC 15442), MM-EO (PA 1-4, PA 6-7, PA 13-19, ATCC 15442). Mean zones of growth inhibition obtained for R-EO ranged from 0 to 23 mm. Table 5 presents the mean diameters of inhibition zones (mm) measured after the treatment of planktonic cells with the volatile fractions of EOs.

Finally, the antibiofilm effect exerted by volatile phases of non-emulsified EOs was determined with the AntiBioVol (antibiofilm activity of volatile compounds) methodology. As can be seen in Table 6, PA 6, PA 17–19 and the ATCC 15442 strains were the most susceptible among all tested strains to volatile EOs. Reductions of biofilm cells viability ranged from 5.40% to 52.99% were obtained for the strains after the exposure to MM-EO. Volatile fractions of 96% (*v*/*v*) ethanol reduced 100.00% of pseudomonal biofilm.

### 2.6. Microscopic Visualization of Biofilm

The R-EO’s high activity against *P. aeruginosa* biofilms was additionally confirmed by fluorescence microscopy (Figure 3). While the high amount of live bacterial, biofilm-forming cells (dyed green) was observed in the untreated control setting (Figure 3A,C), the exposure of biofilm to the liquid (Figure 3B) or the volatile (Figure 3D) R-EO resulted in the high increase in dead/damaged biofilm-forming cells (dyed red/orange).

### 2.7. Statistical Analysis

Statistical analysis was performed to evaluate statistically significant differences between EOs’ antimicrobial activity. A summary of significance levels for each method is presented in Table 7.

## 3. Discussion

EOs are volatile plant derivatives of global medical interest due to their high antimicrobial activity. However, the number of studies focused specifically on the antimicrobial/antibiofilm potential of EOs’ volatile forms is still limited. EO antimicrobial activity depends on a plethora of factors: EO state of matter (volatile or liquid one), chemical composition, hydrophilicity/hydrophobicity, species/strain of microorganism they act against, but also on the type of methodology applied to analyze the aforementioned potential [13]. Therefore, the aim of this paper was to investigate and compare the anti-pseudomonal efficacy of seven EOs in their liquid forms and their volatile fraction with the use of a diversified spectrum of techniques in order to obtain cohesive data. First, the content of the EOs’ constituents was determined to confirm the presence of molecules recognized as those exhibiting antimicrobial potential (Table 1, Appendix A). Next, the pseudomonal strains’ ability to form biofilm was assessed (Figure 1). Having preliminary tests performed, the evaluation of the EOs in their liquid forms’ antimicrobial activity against *P. aeruginosa* planktonic cells was made with the use of two different techniques. Their results indicated that R-EO was the most effective one among the tested EOs (Table 2 and Table 3). In addition, it is suggested that the strong antibacterial potential of R-EO is associated with the activity of its main component, 1,8-cineole [14,15,16]. However, 1,8-cineole is also the major compound of E-EO (Table 1); this specific oil exhibited significantly lower anti-pseudomonal efficacy than R-EO. Other research teams indicated that the antimicrobial effectiveness of crude R-EO and E-EO was stronger than 1,8-cineole applied as a self-reliant antimicrobial agent [17,18]. Therefore, the synergistic (or at least additive) effect of remaining R-EO constituents may also account for the overall oil’s activity against *P. aeruginosa*. It is hypothesized that the smaller the droplets of EO emulsions, the higher the antimicrobial effect that occurs [19]. As we have shown earlier, the droplet diameters of the E-EO emulsion were 2201 nm, while R-EO was 783 nm [20]. The minimal inhibitory concentration (MIC) values (Table 3) of L-EO were 25% (*v*/*v*). MIC values of E-EO were: 12.5% (*v*/*v*) for PA 19 strain and 25% (*v*/*v*) for the rest of the strains, whereas the zones of growth inhibition being the result of antimicrobial activity of L-EO vs. E-EO were 0-7 and 0-18 mm, respectively (Table 2). This interesting observation requires additional experiments to be elucidated because a broad spectrum of possible variables may contribute to the discrepancy between outcomes measured by two testing methods. As an example, the differences in water solubility of the main EO ingredients also have an impact on diffusion through agar. Because agar is mainly composed of water, the higher the aqueous solubility of the compounds, the better their diffusion across the agar. The aqueous solubility of 1,8-cineole, which predominates in E-EO, is 2.63 mg/mL at 293 K, while the solubility of linalool (the main component of L-EO) is 1.34 mg/mL [21]. Linalool diffuses more poorly through the agar than 1,8-cineole, thus, the growth inhibition zones of L-EO are less than E-EO.

To evaluate the antibiofilm efficacy of EOs in their liquid forms, the microdilution method and the modified A.D.A.M. (antibiofilm dressing’s activity measurement) assay were conducted. In the microdilution assay, significant biofilm-forming cells viability reduction was demonstrated for the emulsions of T-EO, TT-EO, B-EO, R-EO and L-EO in their liquid forms at a concentration of 25% (*v*/*v*). In the modified A.D.A.M. methodology, levels of biofilm-forming cells viability reduction were diversified and dependent on the EO applied and on the specific strain exposed to the EO’s activity. Nevertheless, by means of both methods, R-EO, T-EO and TT-EO in their liquid forms can be pointed out as the most potent against the pseudomonal biofilm, which is in line with the results of other research teams. It was reported that above 90% reduction of *P. aeruginosa* biofilm was obtained after the incubation with T-EO and that 1,8-cineole, the main component of R-EO, greatly affected *P. aeruginosa* biofilm formation and disrupted the mature biofilm [18,22,23,24]. The volatile fractions of R-EO and TT-EO also displayed high effectiveness against *P. aeruginosa* biofilm, which may result from their multiple mechanisms of anti-biofilm activity.

In the case of volatile fraction assays, not only does the volatility of the components determine EO activity but also the number and concentration of particular molecules adhered to the agar surface where biofilm forms. In the standard inverted Petri dish method, the EOs are applied to a small stretch (6 mm in diameter paper disc). Therefore, the obtained zones of growth inhibition also depend on the volatiles spreading. The volume of the EOs used in the inverted Petri dish method is lower, and the tightness of the setting is poorer than in the AntiBioVol (antibiofilm activity of volatile compounds) technique; thus, the loss of volatiles is higher.

As mentioned above, volatile fractions of EOs exhibited higher antimicrobial activity than those EOs in their liquid forms in some of the research reports [25]. However, opposite results are reported in the present paper (Table 5 and Table 6). In turn, outcomes of other studies stay in line with the data provided in our work and indicate that EOs in their liquid forms are more active against bacteria than in volatiles, due to the direct contact of molecules with the microorganism [9]. It was also suggested that the antimicrobial activity of EOs’ volatiles is related to the volatility of the EOs’ compounds and their adsorption into the agar surface, which is associated with its hydrophilic character [26,27]. In the previous paper, we demonstrated that the adsorption of EO compounds to the agar changes with the time of exposure [28]. Moreover, the maximum concentration of molecules adsorbed into the agar surface was approximately 40% [28]. Therefore, the working concentration of the active substances is much lower than in the liquid tests. The obtained differences in both fractions’ activity may result from various times of biofilm incubation with the applied indicators of cell metabolic activity (resazurin and tetrazolium chloride). According to the AntiBioVol results, TT-EO, R-EO, MM-EO, and L-EO were the most potent ones among the analyzed EOs. The speed of EOs evaporation is related to their vapor pressure; the higher the vapor pressure, the faster they evaporate. The vapor pressures of EOs main compounds at 25 °C are: thymol, 2.2 Pa; terpinen-4-ol, 6.4 Pa; methyl chavicol, 22.0 Pa; 1,8-cineole, 253.0 Pa; menthol, 19.0 Pa; linalyl acetate, 61.0 Pa. In our study, we found no correlation between the above-mentioned parameter and the level of antibacterial activity of EOs. Such a phenomenon may be explained by the fact that besides vapor pressure, EO’s particular antimicrobial mechanism of action and affinity to agar surface should be considered to analyze the total level of antimicrobial activity. Additionally, strain-dependent tolerance to EOs was observed when AntiBioVol was applied, similar to what was observed when the modified A.D.A.M. method was used. The EOs examined in the current study were also scrutinized in our previous line of investigation toward a Gram-positive bacterium, *S. aureus* [20]. Comparing results from the recent and the present work, it has to be stated that both fractions of all EOs (except for R-EO) displayed higher antimicrobial activity against *S. aureus* than *P. aeruginosa*. It may be related to the fact that, because of the hydrophobic nature, EOs react with lipids of the cell membrane and lead to a leak of intracellular substances and to damage of the cell. Gram-negative bacteria hydrophilic cell walls hinder the penetration of lipophilic EOs, resulting in their higher tolerance to EOs than in the case of Gram-positive pathogens. Furthermore, Gram-negative *P. aeruginosa* is a ubiquitous, opportunistic microorganism, forming a robust biofilm on solid surfaces at the water–air interface [29,30,31]. It thrives there by developing a vast spectrum of adaptive resistance to various antimicrobial agents, probably including those secreted by plants. [32]. Contrary to other EOs, MIC values of R-EO in its liquid emulsified form were equal for *S. aureus* and *P. aeruginosa*. (Table 3) [20]. An explanation of this phenomenon may be the hypothesis that the charge of the bacterial cell surfaces and the cell shape play a role in their tolerance to EOs as well [33,34,35]. Hajlaouli et al. demonstrated a more evident reduction of the bacterial cells’ negative charge for the Gram-negative versus Gram-positive bacteria after the treatment with EOs [34]. Rod-shaped bacterial cells were also more susceptible to EOs than the coccoid ones [35]. In the present study, EOs in their liquid forms and their volatile fractions were investigated because the antimicrobial effect of EOs may alter among different in vitro conditions. R-EO exhibited the highest antibacterial effectiveness against *P. aeruginosa* of all tested EOs. The reported data confirm the high potency of EOs against *P. aeruginosa* biofilms and planktonic cells. Therefore, we are convinced that the data presented in this paper, showing EOs as a promising alternative to the current (performed by means of antibiotics and antiseptics, mostly) anti-pseudomonal therapies, will finally pave the way for novel solutions and approaches aiming to significantly reduce the risk associated with pseudomonal biofilm-related infections.

## 4. Materials and Methods

### 4.1. Bacterial Strains and Culture Conditions

For research purposes, one reference strain of *Pseudomonas aeruginosa* 15442 from the American Type and Culture Collection (ATCC) and fourteen clinical strains (later referred to as PA 1–7 and PA 13–19) of this bacterial species were applied. The strains were part of the Strain and Line Collection of Pharmaceutical Microbiology and Parasitology Department of the Medical University of Wroclaw. The clinical strains were obtained in the year 2016 during the internal Wroclaw Medical University SUB. D198.16.001 project: “The insight into biofilm-related properties of clinical microorganisms and possibilities of their eradication”. All patients provided written consent to participate in the trial and allowed the material obtained during the study (exudate, biopsy specimen, microorganisms) to be used for scientific purposes. The study was approved by the Bioethical Committee of Wroclaw Medical University, protocol # 8/2016.

In each of the performed experiments, the microorganisms’ overnight cultures in the TSB medium (Tryptic Soy Broth, Biomaxima, Lublin, Poland) were prepared, and 0.5 McFarland suspensions were established afterward in a 0.9% (*w/v*) solution of sodium chloride (NaCl, Stanlab, Lublin, Poland) using a densitometer (Densilameter II Erba Lachema, Brno, the Czech Republic).

### 4.2. Essential Oils

Seven commercially available essential oils in their liquid forms and their volatile fractions were tested in the research. The applied EOs are listed in Table 8.

Due to the volatility of EOs, individual essential oils and control settings were examined on separate plates. Moreover, to prevent EO evaporation, in the experiments where their volatile phases were investigated, plates were sealed with parafilm.

### 4.3. GC–MS (Gas Chromatography–Mass Spectrometry) Analysis of the Tested EOs Composition

Essential oils (EOs) were diluted with hexane (JTB, GB), vortexed, and immediately analyzed. All analyses were performed in triplicate.Analysis was carried out using the system Agilent 7890B GC coupled with 7000GC/TQ system connected to PAL RSI85 autosampler (Agilent Technologies, Palo Alto, CA, USA).

The applied column was HP-5 MS; 30 m × 0.25 mm × 0.25 µm (J&W, Agilent Technologies, Palo Alto, CA, USA). Helium was used as a carrier gas at a total flow of 1 mL/min. Chromatographic conditions were applied as follows: split injection in a ratio 100:1, the injector was set on 250 °C, oven temperature program was: 50 °C held for 1 min, then 4 °C/min up to 130 °C, 10 °C/min to 280 °C, and then isothermal for 2 min. The MS detector operated in the electronic impact ionization mode at 70 eV; transfer line, source, and quadrupole temperatures were set at 320, 230, and 150 °C, respectively. Masses were registered in a range from 30 to 400 m/z. Peaks identification was performed using MassHunter Workstation Software Version B.08.00 coupled with the NIST17 mass spectra library and accomplished by comparison with linear retention indexes. The relative abundance of each EO constituent was expressed as percentage content based on the peak area normalization. Regarding the obtained outcomes, only TT-EO met the requirement of pharmacopeial standards. Although, the analysis was not performed in accordance with the normalization procedure from Polish Pharmacopea XI (different column and different temperature program).

### 4.4. Biofilm Biomass Level Assay

The crystal violet assay was performed in 96-well plates (Wuxi Nest Biotechnology, Wuxi, China) to evaluate the total biofilm mass. Briefly, 0.5 McFarland bacterial suspensions, prepared as described in Materials and Methods Section 4.1., were diluted 1000 times in TSB (Tryptic Soy Broth, Biomaxima, Lublin, Poland) medium, and 100 µL was poured into the wells of the plates. The plates were then incubated under static conditions for 24 h at 37 °C. Subsequently, the medium above the biofilm was gently pipetted out, and the plates were dried for 10 min at 37 °C. Then, 100 µL of 20% (*v*/*v*) aqueous crystal violet solution (Aqua-med, Lodz, Poland) was added to each well for 10 min at room temperature. Biofilm cells were rinsed twice with 100 µL of 0.9% (*w/v)* solution of sodium chloride (NaCl, Stanlab, Lublin, Poland) to remove unbounded cells and excess stain. The plates were transferred to the incubator (37 °C) for 10 min. Next, 100 µL of 30% (*v*/*v*) water solution of acetic acid (Chempur, Piekary Slaskie, Poland) was introduced to the wells, and the plates were shaken for 30 min at room temperature at 450 rpm (Mini-shaker PSU-2T, Biosan SIA, Riga, Latvia). The solution was transferred to fresh 96-well plates, and the absorbance was measured at 550 nm using a spectrophotometer (Multiskan Go, Thermo Fisher Scientific, Vantaa, Finland). One independent experiment was performed with six technical replicates.

### 4.5. Biofilm Activity Level Assessment

The presented protocol of biofilm culturing in Materials and Methods Section 4.4. was applied. After the biofilm formation, 10 µL of 0.1% (*w/v*) resazurin sodium salt (7-Hydroxy-3H-phenoxazin-3-one-10-oxide sodium salt, Acros Organics, Geel, Belgium) solution in TSB (Tryptic Soy Broth, Biomaxima, Lublin, Poland) was added to the wells, and the plates were incubated at 37 °C for two hours. The solution was transferred to fresh 96-well plates (Wuxi Nest Biotechnology, Wuxi, China), and its absorbance was measured at 570 and 600 nm using a spectrophotometer (Multiskan Go, Thermo Fisher Scientific, Vantaa, Finland). To assess the cells viability, the absorbance value at 600 nm was subtracted from the value at 570 nm. One independent experiment was carried out with six technical repetitions.

### 4.6. Methods for the Assessment of the Activity of EOs in Their Liquid Forms

#### 4.6.1. The Disc Diffusion Method

In the experiment, 90 mm diameter, 14.2 mm height Petri dishes (Noex, Komorniki, Poland) with 5 mm thick Mueller–Hinton agar layers (Biomaxima, Lublin, Poland) were used. Standard paper discs (diameter of 6 mm, 0.5 mm thickness) were introduced to the wells of 48-well plates (Thermo Fisher Scientific, Waltham, MA, USA), and 0.2 mL of each non-emulsified EOs or saline (control of bacterial growth) (NaCl, Stanlab, Lublin, Poland) was added. The plates were wrapped with tape and kept at 4 °C for 30 min to soak the discs. In the second step of the tests, the Petri dish plates were inoculated with the bacterial suspensions at density 0.5 McFarland prepared as described in Materials and Methods Section 4.1. Next, the soaked paper discs were placed onto the agar layer for assessing the antimicrobial activity of pure EOs in their liquid forms. The dishes were incubated for 24 h at 37 °C, and bacterial growth inhibition zones were measured in mm with a ruler. If no total inhibition was obtained, zones of partial growth inhibition were assessed (in mm). When unequal zones were observed, a shorter diameter was included. One independent experiment with three repetitions was performed, and the mean diameters were calculated.

#### 4.6.2. Assessment of the Minimal Inhibitory Concentrations of EOs Emulsions

To determine the minimal inhibitory concentration (MIC) values of the EOs in their liquid forms, their emulsions in Tween 20 (Zielony Klub, Kielce, Poland) were prepared. In the first step, each EO was combined with the solution of 1.0% (*v*/*v*) Tween 20 in TSB medium (Tryptic Soy Broth, Biomaxima, Lublin, Poland) in ratio 1:1 and mixed with a vortex (Micro-shaker type 326 m, Premed, Marki, Poland) for 30 min at room temperature. Following, geometric dilutions were prepared in TSB and shaken for 30 s. Subsequently, 0.5 MacFarland bacterial suspensions were made according to the description in Materials and Methods Section 4.1. and diluted 1000 times in the TSB medium. Next, 100 µL of the suspensions was added to the wells of 96-well plates (Jet Bio-Filtration Co. Ltd., Guangzhou, China), and the same volume of diluted emulsions was poured. Therefore, the final concentration of each EO applied in the test ranged from 25.0% (*v*/*v*) to 0.01% (*v*/*v*). The following additional samples were included: control of bacterial growth (bacteria with medium), control of medium sterility (medium only), control of 1.0–0.002% (*v*/*v*) Tween 20 antimicrobial activity and samples of emulsions only. The absorbance of the solution was measured at 580 nm using a spectrophotometer (Multiskan Go, Thermo Fisher Scientific, Vantaa, Finland). During the 24 h incubation at 37 °C, the plates were shaken at 350 rpm (Mini-shaker PSU-2T, Biosan SIA, Riga, Latvia). The absorbance was measured immediately after the incubation at the same wavelength. The MIC value was assessed in the concentration (%) (*v*/*v*) in which the difference between the absorbance after and before the sample’s incubation was equal to or lower than zero. Two independent experiments were performed, each with three technical replicates.

#### 4.6.3. Assessment of the Minimal Biofilm Eradication Concentrations of EOs Emulsions

The EOs emulsions and bacterial suspensions applied to assess the Minimal Biofilm Eradication Concentration (MBEC) values were prepared as was elaborated in Materials and Methods Section 4.6.2.. On the first day of the experiment, 100 µL of the aforementioned suspensions was added to the wells of 96-well plates (Jet Bio-Filtration Co. Ltd., Guangzhou, China) containing 100 µL of TSB medium (Tryptic Soy Broth, Biomaxima, Lublin, Poland). The plates were incubated at 37 °C for 24 h under static conditions. Subsequently, the medium was aspirated, and 200 µL of EOs emulsions geometric dilutions were added to the wells. The concentration of each EO applied in the test ranged from 25.0% (*v*/*v*) to 0.01% (*v*/*v*). Control of bacterial growth (bacteria with medium) and medium sterility (medium only) were also prepared. The plates were re-incubated for the next 24 h. Based on the results of the preliminary study (data not shown), the MBEC values of T-EO and TT-EO were determined using TTC (2,3,5-triphenyl-tetrazolium chloride, AppliChem Gmbh, Darmstadt, Germany) as the indicator, while resazurin sodium salt (7-Hydroxy-3H-phenoxazin-3-one-10-oxide sodium salt, Acros Organics, Geel, Belgium) was applied to other EOs. The following steps were executed for the resazurin assay: 40 µL of 0.05% (*w/v*) resazurin solution in TSB medium was added to the biofilm wells treated with EOs emulsions and to the control wells. The plates were incubated for 2 h at 37 °C with continuous shaking at 400 rpm (Mini-shaker PSU-2T, Biosan SIA, Riga, Latvia). The absorbance was measured at 570 and 600 nm using a spectrophotometer (Multiskan Go, Thermo Fisher Scientific, Vantaa, Finland). To calculate the final absorbance of the tested substance, the absorbance value at 600 nm was subtracted from the value at 570 nm. The MBEC values were determined as the lowest concentration of the emulsions where the obtained difference was equal to or lower than zero. For the TTC methodology, 0.2% (*w/v*) TTC solution in TSB was added in the volume of 150 µL to the biofilm after its incubation with EOs emulsions. The plates were incubated for 2 h at 37 °C. Next, the MBEC value was visually determined in the lowest concentration where no red color was observed. Due to the fact that in no case was the MBEC assessed, the three highest concentrations of each EO emulsion for each strain were chosen for further analysis. The contents of the mentioned wells were transferred to 1.5 mL Eppendorf tubes, and 350 µL of 0.1% (*w/v*) solution of saponin (VWR Chemicals, Radnor, PA, USA) was added, and the Eppendorf tubes were vortexed for 1 min. Then, 700 µL of methanol (Stanlab, Lublin, Poland) was added, and the Eppendorf tubes were vortexed for 30 min. Finally, the Eppendorf tubes were centrifuged for 1 min at 3000 rpm, and 100 µL of the supernatant was transferred in three replicates to the wells of the 96-well plates. The same procedure was performed for the control growth wells. The absorbance was measured at 490 nm.

The level of the reduction of biofilm-forming cells viability was calculated according to the formula.
(1)$$\mathrm{Cells}\;\mathrm{viability}\;\mathrm{reduction}\;(\%)=100\%-\left(\frac{\mathrm{AbS}}{\mathrm{AbC}}*100\%\right)$$

AbS, absorbance of the tested substance;

AbC, mean absorbance of growth control.

Two independent experiments were performed, each with three technical replicates.

#### 4.6.4. Evaluation of Antibiofilm Activity of Non-Emulsified EOs Using the Modified A.D.A.M. (Antibiofilm Dressing’s Activity Measurement) Assay

The methodology was a modification of the procedure displayed in our previous research [36]. The following steps of the experiment were performed:

##### Preparation before the Experiment 

To treat the biofilm with the EOs, biocellulose discs (BC) were produced. A *Komagataeibacter xylinus* ATCC 53524 (American Type and Culture Collection) strain was used for this purpose. A Herstin–Schramm (H-S) medium composed of 2% (*w/v*) glucose (Chempur, Piekary Slaskie, Poland), 0.5% (*w/v*) yeast extract (VWR, Radnor, PA, USA), 0.5% (*w/v*) bactopeptone (VWR, Radnor, PA, USA), 0.115% (*w/v*) citric acid monohydric (POCH, Gliwice, Poland), 0.27% (*w/v*) Na2HPO4 (POCH, Gliwice, Poland), 0.05% (*w/v*) MgSO4·7H2O (POCH, Gliwice, Poland), and 1% (*v*/*v*) ethanol (Chempur, Piekary Slaskie, Poland) was used for bacterial culturing. Then, 1 mL of H-S medium was added to the wells of a 24-well plate (Wuxi Nest Biotechnology, Wuxi, China) and inoculated with *K.xylinus*. The plate was incubated for 7 days at 28 °C to obtain 14 mm BC discs. Then, the discs were removed and washed with 0.1 M NaOH (Chempur, Piekary Slaskie, Poland) at 80 °C. Next, BC discs were washed with double-distilled water to neutralize their pH and autoclaved. To evaluate the concentration of substances adsorbed into the BC discs, six of them were weighed, dried at 37 °C and weighed again. The average concentration (%) of the adsorbed liquid was calculated with the formula:Compound concentration (%) = [EV/((WBC − DBC) + EV)] ∗ 100%(2)

EV, a volume of essential oil (mL);

WBC, the weight of wet BC disc (g);

DBC, the weight of dry BC disc (g).

##### First Day of the Experiment

In the first line of the investigation, 1.5 mL of Brain Heart Infusion Broth (BHI, Biomaxima, Lublin, Poland) with 2% (*w/v*) of Bacteriological Lab Agar (Biomaxima, Lublin, Poland) was poured into the wells of a 24-well plate (further referred to as Agar Plate). After the agar solidifies, 8 mm in diameter plugs were cut out of each well using a cork-borer. The plugs were removed and discarded to make 8 mm in diameter tunnels in agar, and the Agar Plate was kept refrigerated for the next day. Simultaneously, the same agar formulation was used to prepare a 6 mm in height agar layer on a Petri dish (Noex, Komorniki, Poland). Then, agar plugs 8 mm in diameter were cut out from the agar Petri dish and placed in a fresh 24-well plate (further referred to as Plugs Plate). The microorganisms’ suspensions, prepared according to the description in Materials and Methods Section 4.1., were diluted 1000 times in TSB medium (Tryptic Soy Broth, Biomaxima, Lublin, Poland), introduced to the wells of the Plugs Plate in the volume of 2 mL and transferred to the incubator (37 °C) for biofilm formation. The BC discs, prepared as described above, were placed in a fresh 24-well plate (further referred to as Discs Plate), and 1 mL of undiluted, non-emulsified essential oils or saline (control of bacterial growth), or 96% (*v*/*v*) ethanol (Chempur, Piekary Slaskie, Poland) was added, and the plate was sealed with parafilm and placed at 4 °C. Both plates (Plugs Plate 2 and Discs Plate) were incubated for 24 h.

##### Second Day of the Experiment

Subsequently, biofilm plugs were taken out from the Plugs Plate and placed in the bottom of agar tunnels in the Agar Plate so that the biofilm was on the top of the plugs. Then, 50 µL of TSB medium was added to fill up the space in the tunnels. The BC discs soaked with non-emulsified EOs/saline/ethanol were transferred from the Disc Plate and placed on the top of the agar wells in the Agar Plate. The biofilm was adhered to the plug and had no direct contact with the BC. Substances were gradually released from the BC to the medium. The experimental setting was incubated for 24 h/ 37 °C under static conditions.

##### Third Day of the Experiment

After the biofilm treatment with the tested substances, the BC discs were removed. The content of each agar tunnel (medium and agar plugs) was transferred to the wells of 48-well plates (Wuxi Nest Biotechnology, Wuxi, China) and 1 mL of 0.002% (*w/v*) resazurin sodium salt (7-Hydroxy-3H-phenoxazin-3-one-10-oxide sodium salt, Acros Organics, Geel, Belgium) in TSB was added. The plates were shaken at 350 rpm (Mini-shaker PSU-2T, Biosan SIA, Riga, Latvia) for 4 h and 15 min at 37 °C. Then, 100 µL of the color solution was transferred to 96-well plates (Jet Bio-Filtration Co. Ltd., Guangzhou, China) in three replications from each 48-well. The absorbance was measured at 570 and 600 nm using a spectrophotometer (Multiskan Go, Thermo Fisher Scientific, Vantaa, Finland). To calculate the final absorbance of the tested substance, the absorbance at 600 nm was subtracted from the value at 570 nm. The formula calculated the reduction of cells viability:(3)$$\mathrm{Cells}\;\mathrm{viability}\;\mathrm{reduction}\;(\%)=100\%-\left(\frac{\mathrm{AbS}}{\mathrm{AbC}}*100\%\right)$$

AbS, absorbance of the tested substance;

AbC, mean absorbance of growth control.

One independent experiment was performed with six technical replicates. Antibiofilm activity of ethanol was examined only against the *P. aeruginosa* 15442 strain.

### 4.7. Methods for the Assessment of EOs’ Volatile Fractions Activity

#### 4.7.1. The Inverted Petri Dish Methodology

The assay was performed similarly to the method presented in Materials and Methods Section 4.6.1. The paper disc soaked with the non-emulsified tested substances was solely placed onto the lid of the Petri dish.

#### 4.7.2. Evaluation of Antibiofilm Activity of Non-Emulsified EOs Using the AntiBioVol (Antibiofilm Activity of Volatile Compounds) Method

The assay was performed based on the protocol presented in our previous study [37].

##### First Day of the Experiment

In this part of the investigation, the Agar Plate, the Petri dish with the plugs and the Plugs Plate were prepared as mentioned in Materials and Methods Section 4.6.4. The following modifications were made. First, wells of the Agar Plate were filled with BHI (Brain Heart Infusion Broth, Biomaxima, Lublin, Poland) and agar to full. Second, twice as many agar plugs were cut out from the Petri dish, and part of them remained sterile. They were placed on the bottom of the agar tunnels (Agar Plate) and kept refrigerated until the next day.

##### Second Day of the Experiment

Biofilm plugs were taken out from the Plugs Plate and placed in the agar tunnels of the Agar Plate on the top of the sterile ones. Then, 0.5 mL of undiluted, non-emulsified EOs or saline (control of growth) or ethanol (Chempur, Piekary Slaskie, Poland) was added to the wells of a fresh 24-well plate (later referred to as Substance Plate) (Wuxi Nest Biotechnology, Wuxi, China). The Agar Plate was placed upside down on the Substance Plate, and the agar wells were set directly above the wells with the tested substances. The plugs’ diameters were the same as the tunnels in which they were placed, and plates were gently transferred to protect the plugs from dropping into the wells of the Substance Plate. The plates were sealed and incubated for 24 h at 37 °C under static conditions.

##### Third Day of the Experiment

As the incubation was completed, the plates were separated, and the upper plugs (containing biofilms) were transferred to fresh 48-well plates (Wuxi Nest Biotechnology, Wuxi, China), and 1 mL of 0.002% (*w/v*) resazurin sodium salt (7-Hydroxy-3H-phenoxazin-3-one-10-oxide sodium salt, Acros Organics, Geel, Belgium) in TSB was poured. The incubation was continued for 2 h/ 37 °C with shaking at 350 rpm. Then, 100 µL of the color solution was transferred to 96-well plates (Jet Bio-Filtration Co. Ltd Guangzhou, China)) in three replications from each 48-well. The absorbance was measured at 570 and 600 nm using a spectrophotometer (Multiskan Go, Thermo Fisher Scientific, Vantaa, Finland). The absorbance at 600 nm was subtracted from the value at 570 nm. The reduction of cells viability was calculated by the formula:(4)$$\mathrm{Cells}\;\mathrm{viability}\;\mathrm{reduction}\;(\%)=100\%-\left(\frac{\mathrm{AbS}}{\mathrm{AbC}}*100\%\right)$$

AbS, absorbance of the tested substance;

AbC, mean absorbance of growth control.

One independent experiment was performed with six technical replicates. Antibiofilm activity of ethanol was examined only against the *P. aeruginosa* 15442 strain.

### 4.8. Microscopic Visualization of Biofilm

The pseudomonal biofilm (ATCC 15442 strain) treated with R-EO or saline by means of the A.D.A.M. (antibiofilm dressing’s activity measurement) or the AntiBioVol (antibiofilm activity of volatile compounds) technique (as presented in Materials and Methods Section 4.6.4. or Section 4.7.2, respectively) was immersed in 1 mL of Filmtracer™ LIVE/DEAD™ Biofilm Viability Kit (Invitrogen, Thermo Fisher Scientific, USA) solution and incubated at room temperature for 15 min. After incubation, the solution was removed, and the biofilms were gently rinsed once with sterile water. The biofilms were analyzed using a confocal microscope (Leica, SP8, Wetzlar, Germany) with a 25× water dipping objective, using sequential mode for 488 nm laser line and 500–530 nm emission to detect SYTO-9 and 552 nm laser line and 575–627 nm emission to detect propidium iodide (PI) within microbial cells.

### 4.9. Statistical Analysis

Statistical analysis was performed using Statistica (Version 13; TIBCO Software Inc., Palo Alto, CA, USA). Normality distribution was assessed with the Shapiro–Wilk test. To compare EOs’ efficacy, a non-parametric ANOVA Kruskal–Wallis test and post hoc Dunn’s analysis were performed. Results with a significance level *p* < 0.05 were considered significant. The statistical analysis is presented in Table 7.

## 5. Conclusions

The antimicrobial activity of EOs’ volatile fractions varies from that of EOs in their liquid forms. The antimicrobial effectiveness of EOs may depend on the type of methodology applied to analyze the antimicrobial activity. Rosemary and tea tree EOs in their liquid forms and their volatile fractions displayed significant antibiofilm effectiveness against *P. aeruginosa*. Volatile fractions of menthol mint EO exhibited the most potent antibiofilm effect among all tested EOs.

## Figures and Tables

**Figure 1 molecules-27-04096-f001:**
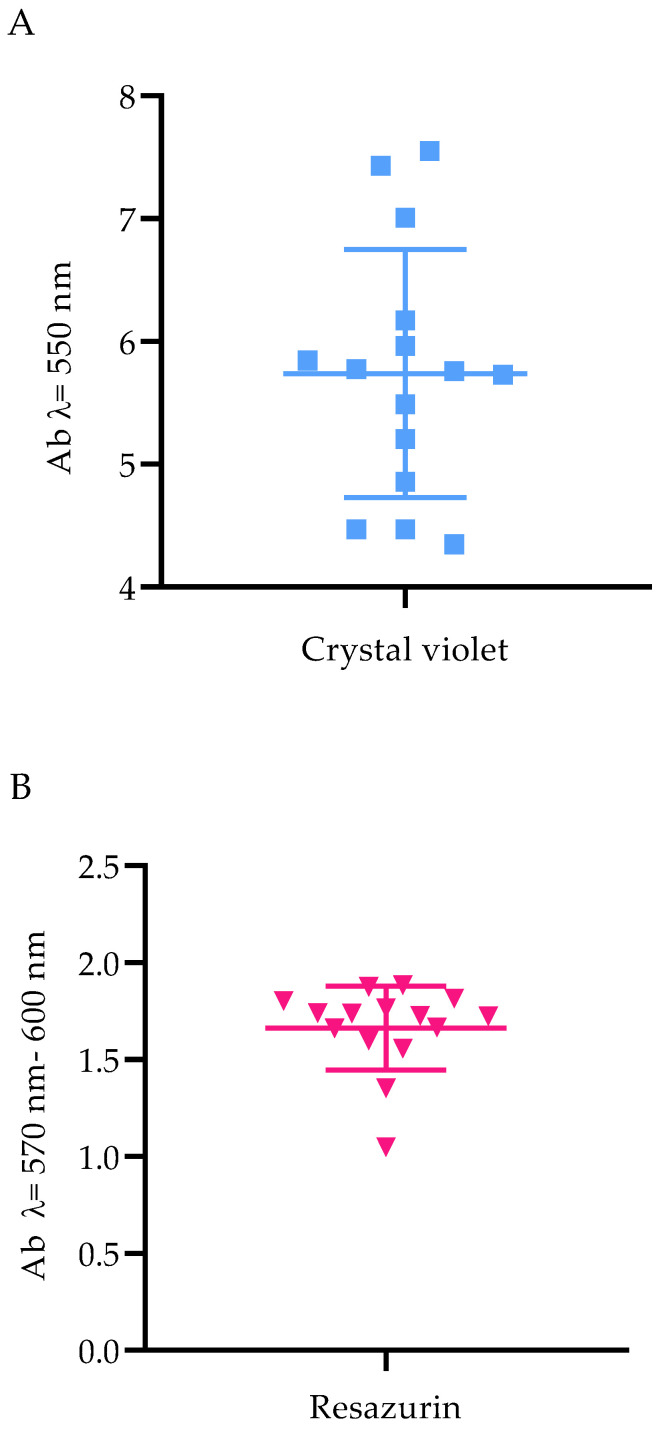
The ability of *Pseudomonas aeruginosa* clinical (PA 1–7, PA 13–19) and the reference (ATCC 15442) strains to form biofilm. (**A**) Biofilm biomass level assessed with the crystal violet method. (**B**) Metabolic activity of biofilm-forming cells, determined with resazurin staining. Ab, absorbance. The average and standard deviations are marked.

**Figure 2 molecules-27-04096-f002:**
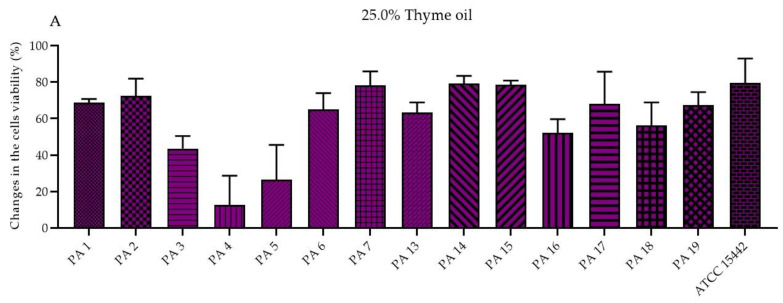
Changes in the biofilm-forming cells viability (%) of *P. aeruginosa* clinical (PA 1–7, PA 13–19) and the reference (ATCC 15442) strains after treatment with emulsified essential oils in their liquid forms in the concentration of 25.0% (*v*/*v*). Results of microdilution methodology with (**A**,**B**) TTC and (**C**–**G**) resazurin staining. Standard deviations are marked. The negative values indicate an increase in biofilm-forming cells viability after their treatment with EOs in comparison to the growth control (untreated cells).

**Figure 3 molecules-27-04096-f003:**
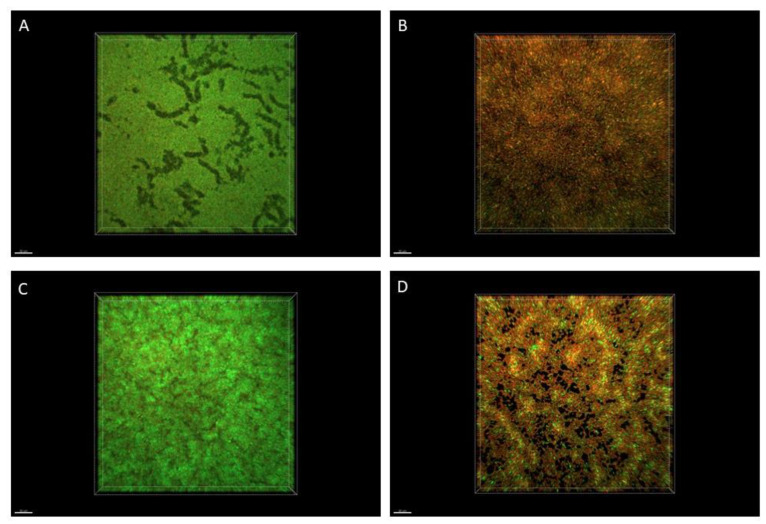
Impact of R-EO on *P. aeruginosa* ATCC 15442 biofilm. (**A**,**B**) Pseudomonal biofilm untreated and treated with R-EO in its liquid form, assessed with the modified A.D.A.M. (antibiofilm dressing’s activity measurement) method. (**C**,**D**) Pseudomonal biofilm untreated and treated with R-EO volatiles, assessed with the AntiBioVol (antibiofilm activity of volatile compounds) assay. The red/orange color shows pseudomonal cells altered/damaged as the result of exposure to R-EO, while green-colored cells are non-altered, viable cells. Moreover, the darker (less green) picture shows that fewer live cells are captured in this particular field of vision.

**Table 1 molecules-27-04096-t001:** The content (% ± standard deviation) of compounds in essential oils. T-EO, thyme oil; TT-EO, tea tree oil; B-EO, basil oil; R-EO, rosemary oil; E-EO, eucalyptus oil; MM-EO, menthol mint oil; L-EO, lavender oil. Dashes (-) indicate the compounds not presented in the specific EO. The components in line with Polish Pharmacopoeia XI standards are marked in green color.

Retention Time (min)	Compound	Mean Concentration (%) ± SD						
		T-EO	TT-EO	B-EO	R-EO	E-EO	MM-EO	L-EO
6.37	α-Thujene	-	1.11 ± 0.02	-	-	-	-	-
6.56	α-Pinene	2.20 ± 0.09	2.85 ± 0.06	-	2.58 ± 0.28	2.30 ± 0.04	3.01 ± 0.17	0.94 ± 0.04
6.98	Camphene	0.73 ± 0.04	-	-	-	-	-	-
7.8	Sabinene	0.64 ± 0.03	0.75 ± 0.02	-	-	-	-	-
8.22	Cyclofenchene	-	-	-	4.99 ± 0.48	-	2.68 ± 0.17	3.66 ± 0.35
8.25	β-Pinene	0.81 ± 0.05	0.69 ± 0.01	-	-	1.09 ± 0.02	-	-
8.65	α-Phellandrene	-	0.61 ± 0.01	-	0.23 ± 0.07	2.02 ± 0.02	-	-
8.69	2-Bornene	-	-	-	3.00 ± 0.29	-	-	-
9.06	α-Terpinene	-	11.07 ± 0.17	-	0.80 ± 0.04	-	-	-
9.31	p-Cymene	26.91 ± 0.99	4.69 ± 0.07	-	-	6.89 ± 0.07	-	-
9.45	Limonene	0.77 ± 0.04	2.08 ± 0.05	-	14.26 ± 0.99	-	3.78 ± 0.27	-
9.52	1.8-Cineole	-	3.34 ± 0.06	-	30.12 ± 1.74	79.10 ± 0.61	-	2.56 ± 0.39
9.57	β -Thujene	-	-	-	2.33 ± 0.11	-	-	-
10.22	Myrcene	-	-	-	0.38 ± 0.14	-	-	-
10.22	Myrcene	-	-	-	-	-	-	0.86 ± 0.13
10.47	γ-Terpinene	8.60 ± 0.03	19.07 ± 0.27	-	8.21 ± 0.35	8.16 ± 0.07	-	-
10.83	3-Carene	-	-	-	-	-	-	1.19 ± 0.12
11.35	o-Cymene	-	-	-	3.15 ± 1.45	-	-	1.19 ± 0.12
11.44	α-Terpinolene	-	4.34 ± 0.06	-	-	-	-	-
11.85	Linalool	3.45 ± 0.15	-	10.69 ± 1.13	-	-	-	37.76 ± 1.18
13.36	Camphor	0.66 ± 0.06	-	-	21.97 ± 0.77	-	-	0.84 ± 0.10
14.58	Terpinen-4-ol	-	33.27 ± 0.79	-	-	-	-	3.03 ± 0.46
14.99	α-Terpineol	7.84 ± 0.30	3.26 ± 0.13	-	1.56 ± 0.39	-	-	-
15.98	Menthone	-	-	-	-	-	24.53 ± 0.23	-
16.08	Isoborneol	-	-	-	1.53 ± 0.06	-	-	-
16.36	Isomenthone	-	-	-	-	-	13.54 ± 1.75	-
16.40	Borneol	-	-	-	2.69 ± 0.09	-	-	2.04 ± 0.56
16.66	Menthol	-	-	-	-	-	45.57 ± 2.21	-
17.59	Methyl chavicol	-	-	89.31 ± 1.13	-	-	-	-
18.49	Thymol	44.00 ± 0.46	-	-	-	-	-	-
19.60	Linalyl acetate	-	-	-	-	-	-	41.13 ± 0.40
20.61	Bornyl acetate	-	-	-	1.17 ± 0.09	-	-	-
20.81	Lavandulyl acetate	-	-	-	-	-	-	1.80 ± 0.22
20.89	Menthyl acetate	-	-	-	-	-	5.61 ± 0.36	-
22.36	β-Caryophyllene	1.00 ± 0.05	0.53 ± 0.01	-	-	-	-	-
22.83	Aromadendrene	-	1.83 ± 0.03	-	-	-	-	-
23.32	Alloaromadendrene	-	0.82 ± 0.02	-	-	-	-	-
24.03	Viridiflorene	-	2.35 ± 0.04	-	-	-	-	-
24.55	β-Cadinene	-	2.78 ± 0.03	-	-	-	-	-
24.80	Caryophyllene	-	-	-	0.85 ± 0.21	-	1.27 ± 0.57	3.47 ± 0.66

**Table 2 molecules-27-04096-t002:** Mean diameters of inhibition zones (mm ± standard deviation) after treatment of *P. aeruginosa* clinical (PA 1–7, PA 13–19) and the reference (ATCC 15442) strains with non-emulsified EOs in their liquid forms assessed with the disc diffusion method. T-EO, thyme oil; TT-EO, tea tree oil; B-EO, basil oil; R-EO, rosemary oil; E-EO, eucalyptus oil; MM-EO, menthol mint oil; L-EO, lavender oil. According to their susceptibility to a particular oil, the strains were divided into two groups of seven or eight samples per group. In the case of B-EO, MM-EO, and L-EO, the zone equal to 0 mm was the parameter for the low-susceptibility group and higher than 0 mm for the high-susceptibility one. The groups are marked as follows: red, low susceptibility among the tested strains; green, high susceptibility among the tested strains.

Mean Zones of Growth Inhibition (mm ± SD) after Treatment with Non-Emulsified EOs in Their Liquid Forms							
Strain	T-EO	TT-EO	B-EO	R-EO	E-EO	MM-EO	L-EO
PA 1	9 ± 1	8 ± 1	10 ± 2	18 ± 5	18 ± 1	8 ± 7	7 ± 1
PA 2	13 ± 3	0 ± 0	12 ± 1	17 ± 5	18 ± 2	0 ± 0	0 ± 0
PA 3	3 ± 5	13 ± 1	0 ± 0	15 ± 4	11 ± 1	0 ± 0	0 ± 0
PA 4	9 ± 1	0 ± 0	0 ± 0	29 ± 2	0 ± 0	0 ± 0	0 ± 0
PA 5	0 ± 0	0 ± 0	0 ± 0	20 ± 2	8 ± 2	6 ± 6	0 ± 0
PA 6	5 ± 5	6 ± 5	0 ± 0	20 ± 1	9 ± 1	7 ± 6	0 ± 0
PA 7	8 ± 7	0 ± 0	0 ± 0	19 ± 1	0 ± 0	11 ± 2	0 ± 0
PA 13	13 ± 4	4 ± 8	10 ± 1	15 ± 4	14 ± 1	0 ± 0	0 ± 0
PA 14	7 ± 1	0 ± 0	0 ± 0	15 ± 1	0 ± 0	0 ± 0	0 ± 0
PA 15	8 ± 0	0 ± 0	8 ± 1	14 ± 1	14 ± 1	0 ± 0	0 ± 0
PA 16	3 ± 6	9 ± 1	0 ± 0	19 ± 0	0 ± 0	0 ± 0	0 ± 0
PA 17	11 ± 1	9 ± 1	0 ± 0	19 ± 1	10 ± 8	0 ± 0	0 ± 0
PA 18	10 ± 1	14 ± 1	0 ± 0	15 ± 0	0 ± 0	4 ± 7	0 ± 0
PA 19	0 ± 0	0 ± 0	0 ± 0	13 ± 2	11 ± 0	0 ± 0	0 ± 0
ATCC 15442	7 ± 0	5 ± 4	0 ± 0	12 ± 0	0 ± 0	0 ± 0	0 ± 0

**Table 3 molecules-27-04096-t003:** Minimal inhibitory concentration (%) (*v*/*v*) of emulsified EOs in their liquid forms against *P. aeruginosa* clinical (PA 1–7, PA 13–19) and the reference (ATCC 15442) strains assessed with the microdilution method. T-EO, thyme oil; TT-EO, tea tree oil; B-EO, basil oil; R-EO, rosemary oil; E-EO, eucalyptus oil; MM-EO, menthol mint oil; L-EO, lavender oil. R symbols indicate EOs where the minimal inhibitory concentration values were not reached in the highest concentration (25.0% (*v*/*v*)) of applied emulsions.

Minimal Inhibitory Concentration (%) of Emulsified EOs in Their Liquid Forms							
Strain	T-EO	TT-EO	B-EO	R-EO	E-EO	MM-EO	L-EO
PA 1	6.3	12.5	6.3	0.4	25.0	25.0	25.0
PA 2	25.0	25.0	12.5	0.8	25.0	25.0	25.0
PA 3	25.0	12.5	3.1	0.4	25.0	25.0	25.0
PA 4	25.0	12.5	6.3	0.4	25.0	25.0	25.0
PA 5	12.5	12.5	3.1	0.4	25.0	25.0	25.0
PA 6	25.0	25.0	6.3	0.8	25.0	25.0	25.0
PA 7	0.2	12.5	1.6	6.3	25.0	R	25.0
PA 13	25.0	12.5	1.6	0.8	25.0	R	25.0
PA 14	25.0	25.0	3.1	0.4	25.0	25.0	25.0
PA 15	25.0	25.0	3.1	0.8	25.0	25.0	25.0
PA 16	R	25.0	12.5	0.8	25.0	25.0	25.0
PA 17	R	25.0	12.5	0.8	25.0	25.0	25.0
PA 18	25.0	25.0	3.1	0.8	25.0	12.5	25.0
PA 19	12.5	25.0	3.1	0.4	12.5	6.3	25.0
ATCC 15442	R	25.0	12.5	0.4	25.0	R	25.0

**Table 4 molecules-27-04096-t004:** Changes in the biofilm-forming cells viability (%) of *P. aeruginosa* clinical (PA 1–7, PA 13–19) and the reference (ATCC 15442) strains after treatment with non-emulsified essential oils in their liquid forms assessed with the A.D.A.M. (antibiofilm dressing’s activity measurement) method. T-EO, thyme oil; TT-EO, tea tree oil; B-EO, basil oil; R-EO, rosemary oil; E-EO, eucalyptus oil; MM-EO, menthol mint oil; L-EO, lavender oil. The strains were grouped by their susceptibility to the particular oil. The groups are marked as follows: red, lowest susceptibility; purple, moderate susceptibility; green, the highest susceptibility among the tested strains. The negative values indicate an increase in biofilm-forming cells viability after their treatment with EOs in comparison to the growth control (untreated cells).

Changes in the Biofilm-Forming Cells Viability (%) after Treatment with Non-Emulsified EOs in Their Liquid Forms							
Strain	T-EO	TT-EO	B-EO	R-EO	E-EO	MM-EO	L-EO
PA 1	47.23	50.57	3.49	69.55	33.14	26.11	43.93
PA 2	52.79	54.53	32.19	52.64	38.57	35.23	38.61
PA 3	63.13	37.74	22.23	87.56	64.39	79.78	69.11
PA 4	38.37	58.28	10.05	81.75	52.14	27.31	61.08
PA 5	69.45	50.36	37.78	70.46	19.31	53.54	54.64
PA 6	60.11	60.23	25.06	37.05	29.49	72.11	60.74
PA 7	36.34	44.24	20.36	38.93	20.23	64.88	40.90
PA 13	51.79	44.36	−34.87	55.01	−16.98	−13.07	20.92
PA 14	49.02	81.27	16.06	31.39	31.71	25.78	27.83
PA 15	63.80	85.39	25.56	36.98	23.81	44.93	31.14
PA 16	77.03	80.88	37.35	53.77	21.97	−12.46	50.29
PA 17	13.07	44.30	36.54	31.32	44.17	43.41	51.57
PA 18	−49.39	−39.67	−138.15	−152.19	−168.89	−110.24	−127.17
PA 19	56.21	83.26	4.13	17.92	1.78	46.16	34.75
ATCC 15442	41.11	47.06	12.50	14.56	11.33	49.87	57.36

**Table 5 molecules-27-04096-t005:** Mean diameters of inhibition zones (mm ± standard deviation) after treatment of planktonic forms of *P. aeruginosa* clinical (PA 1–7, PA 13–19) and the reference (ATCC 15442) strains with volatile fractions of non-emulsified EOs assessed with the inverted Petri dish method. T-EO, thyme oil; TT-EO, tea tree oil; B-EO, basil oil; R-EO, rosemary oil; E-EO, eucalyptus oil; MM-EO, menthol mint oil; L-EO, lavender oil. According to their susceptibility to R-EO, the strains were divided into two groups for seven or eight samples per group. For the rest of the EOs, the zone equal to 0 mm was the parameter for the low susceptibility group and higher than 0 mm for the high susceptibility one. The groups are marked as follows: red, low susceptibility among the tested strains; green, high susceptibility among the tested strains.

Mean Zones of Growth Inhibition (mm ± SD) after Treatment with Volatile Fractions of Non-Emulsified EOs							
Strain	T-EO	TT-EO	B-EO	R-EO	E-EO	MM-EO	L-EO
PA 1	0 ± 0	0 ± 0	0 ± 0	18 ± 15	22 ± 2	0 ± 0	0 ± 0
PA 2	0 ± 0	0 ± 0	0 ± 0	13 ± 13	16 ± 2	0 ± 0	0 ± 0
PA 3	0 ± 0	0 ± 0	0 ± 0	13 ± 6	16 ± 2	0 ± 0	0 ± 0
PA 4	0 ± 0	0 ± 0	0 ± 0	23 ± 2	0 ± 0	0 ± 0	0 ± 0
PA 5	0 ± 0	0 ± 0	0 ± 0	19 ± 3	0 ± 0	3 ± 6	0 ± 0
PA 6	0 ± 0	0 ± 0	0 ± 0	15 ± 2	0 ± 0	0 ± 0	0 ± 0
PA 7	3 ± 5	0 ± 0	0 ± 0	15 ± 3	0 ± 0	0 ± 0	0 ± 0
PA 13	0 ± 0	0 ± 0	0 ± 0	3 ± 6	3 ± 5	0 ± 0	0 ± 0
PA 14	0 ± 0	0 ± 0	0 ± 0	0 ± 0	0 ± 0	0 ± 0	0 ± 0
PA 15	0 ± 0	0 ± 0	0 ± 0	5 ± 4	0 ± 0	0 ± 0	0 ± 0
PA 16	0 ± 0	0 ± 0	0 ± 0	12 ± 4	0 ± 0	0 ± 0	0 ± 0
PA 17	0 ± 0	0 ± 0	0 ± 0	16 ± 5	0 ± 0	0 ± 0	0 ± 0
PA 18	0 ± 0	7 ± 6	0 ± 0	14 ± 6	0 ± 0	0 ± 0	0 ± 0
PA 19	0 ± 0	0 ± 0	0 ± 0	3 ± 5	0 ± 0	0 ± 0	0 ± 0
ATCC 15442	0 ± 0	0 ± 0	0 ± 0	0 ± 0	0 ± 0	0 ± 0	0 ± 0

**Table 6 molecules-27-04096-t006:** Changes in biofilm-forming cells viability (%) of *P. aeruginosa* clinical (PA 1–7, PA 13–19) and the reference (ATCC 15442) strains after treatment with volatile non-emulsified EOs assessed with the AntiBioVol (antibiofilm activity of volatile compounds) method. T-EO, thyme oil; TT-EO, tea tree oil; B-EO, basil oil; R-EO, rosemary oil; E-EO, eucalyptus oil; MM-EO, menthol mint oil; L-EO, lavender oil. The strains were grouped by their susceptibility to the particular oil. The groups are marked as follows: red, lowest susceptibility; purple, moderate susceptibility; green, highest susceptibility among the tested strains. The negative values indicate an increase in biofilm-forming cells viability after their treatment with EOs in comparison to the growth control (untreated cells).

Changes in the Biofilm-Forming Cells Viability (%) after Treatment with Volatile Fractions of Non-Emulsified EOs							
Strain	T-EO	TT-EO	B-EO	R-EO	E-EO	MM-EO	L-EO
PA 1	18.13	11.08	13.23	2.65	13.21	26.86	4.49
PA 2	29.21	24.08	6.18	−0.11	22.29	40.82	2.53
PA 3	10.29	19.17	10.73	−17.52	3.75	25.85	−0.78
PA 4	12.94	14.66	−1.71	−28.46	6.75	33.56	−6.11
PA 5	1.71	50.34	15.55	45.10	18.26	9.98	34.20
PA 6	9.22	53.81	−12.83	37.88	19.24	52.99	32.67
PA 7	3.51	20.04	−83.01	12.43	−96.64	5.40	−81.17
PA 13	8.11	−1.63	22.75	−4.50	3.69	17.62	−2.68
PA 14	12.30	−3.61	−17.21	−5.96	−8.91	19.66	−4.45
PA 15	15.10	−12.49	45.13	−10.29	−1.21	37.22	−3.83
PA 16	18.57	0.17	−3.85	−3.40	0.01	9.49	1.27
PA 17	6.38	22.34	20,84	14.73	13.74	11.22	42.97
PA 18	9.12	27.69	33.45	16.28	12.29	15.00	33.00
PA 19	3.58	27.20	7.97	18.32	11.94	19.00	40.84
ATCC 15442	13.08	53.11	22.80	43.74	−7.53	46.76	−1.71

**Table 7 molecules-27-04096-t007:** Significance levels of differences in changes in pseudomonal biofilm cells viability after treatment with EOs in their liquid forms and volatile fractions obtained with three methods. The differences were statistically significant for *p* < 0.05 and are referred to as *p* < 0.03 (*), *p* < 0.006 (**), *p* < 0.00003 (***); ns refers to difference being statistically insignificant. T-EO, thyme oil; TT-EO, tea tree oil; B-EO, basil oil; R-EO, rosemary oil; E-EO, eucalyptus oil; MM-EO, menthol mint oil; L-EO, lavender oil.

**Comparison of the Changes in the Biofilm-Forming Cells Viability after Treatment with Emulsified EOs in Their Liquid Forms in the Concentration of 25.0% (*v*/*v*)**							
	**T-EO**	**TT-EO**	**B-EO**	**R-EO**	**E-EO**	**MM-EO**	**L-EO**
T-EO	-	ns	***	***	ns	ns	ns
TT-EO	ns	-	**	***	*	**	ns
B-EO	***	**	-	ns	***	***	***
R-EO	***	***	ns	-	***	***	***
E-EO	ns	*	***	***	-	ns	ns
MM-EO	ns	**	***	***	ns	-	ns
L-EO	ns	ns	***	***	ns	ns	-
**Comparison of the Changes in the Biofilm-Forming Cells Viability after Treatment with Emulsified EOs in Their Liquid Forms in the Concentration of 12.5 (*v*/*v*)**							
	**T-EO**	**TT-EO**	**B-EO**	**R-EO**	**E-EO**	**MM-EO**	**L-EO**
T-EO	-	ns	ns	ns	***	***	***
TT-EO	ns	-	ns	ns	***	***	***
B-EO	ns	ns	-	ns	***	***	***
R-EO	ns	ns	ns	-	***	***	**
E-EO	***	***	***	***	-	ns	ns
MM-EO	***	***	***	***	ns	-	ns
L-EO	***	***	***	**	ns	ns	-
**Comparison of the Changes in the Biofilm-Forming Cells Viability after Treatment with Emulsified EOs in Their Liquid Forms in the Concentration of 6.3 (*v*/*v*)**							
	**T-EO**	**TT-EO**	**B-EO**	**R-EO**	**E-EO**	**MM-EO**	**L-EO**
T-EO	-	ns	***	***	***	***	***
TT-EO	ns	-	***	**	***	***	***
B-EO	***	***	-	ns	ns	ns	ns
R-EO	***	**	ns	-	***	ns	*
E-EO	***	***	ns	***	-	ns	ns
MM-EO	***	***	ns	ns	ns	-	ns
L-EO	***	***	ns	*	ns	ns	-
**Comparison of the Changes in the Biofilm-Forming Cells Viability after Treatment with Non-Emulsified EOs in Their Liquid Forms**							
	**T-EO**	**TT-EO**	**B-EO**	**R-EO**	**E-EO**	**MM-EO**	**L-EO**
T-EO	-	ns	***	ns	**	ns	ns
TT-EO	ns	-	***	ns	***	*	ns
B-EO	***	***	-	***	ns	**	***
R-EO	ns	ns	***	-	**	ns	ns
E-EO	**	***	ns	**	-	ns	*
MM-EO	ns	*	**	ns	ns	-	ns
L-EO	ns	ns	***	ns	*	ns	-
**Comparison of the Changes in the Biofilm-Forming Cells Viability after Treatment with Volatile Fractions of Non-Emulsified EOs**							
	**T-EO**	**TT-EO**	**B-EO**	**R-EO**	**E-EO**	**MM-EO**	**L-EO**
T-EO	-	ns	ns	ns	ns	**	ns
TT-EO	ns	-	ns	**	**	ns	**
B-EO	ns	ns	-	ns	ns	**	ns
R-EO	ns	**	ns	-	ns	***	ns
E-EO	ns	**	ns	ns	-	***	ns
MM-EO	**	ns	**	***	***	-	***
L-EO	ns	**	ns	ns	ns	***	-

**Table 8 molecules-27-04096-t008:** List of the essential oils analyzed in the paper.

Common Name of EO	Plant Origin	Part of the Plant	Abbreviation	Manufacturer, City, Country
thyme oil	*Thymus vulgaris* L.	herb	T-EO	Etja, Elblag, Poland
tea tree oil	*Melaleuca alternifolia* Cheel.	leaves	TT-EO	Pharmatech, Zukowo, Poland
basil oil	*Ocimum basilicum* L.	flowers	B-EO	Nanga, Zlotow, Poland
rosemary oil	*Rosmarinus officinalis* L.	flowering shoots	R-EO	Nanga, Zlotow, Poland
eucalyptus oil	*Eucalyptus globulus* Labill.	leaves and twigs	E-EO	Pharmatech, Zukowo, Poland
lavender oil	*Lavandula angustifolia* Mill.	flowering herb	L-EO	Kej, Cirkowice, Poland
menthol mint oil	*Mentha arvensis* L.	leaves	MM-EO	Optima Natura, Grodki, Poland

## Data Availability

All necessary data are presented in the manuscript and in the supplementary data and can be provided from the authors upon reasonable request.

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
