# Peer review of "Chemical Composition and Antibacterial Activity of Liquid and Volatile Phase of Essential Oils against Planktonic and Biofilm-Forming Cells of Pseudomonas aeruginosa"

_molecules, 2022, doi:10.3390/molecules27134096_

Round 1

Reviewer 1 Report

Dear authors,

The manuscript titled: “The comparison between in vitro activity of volatile and liquid fractions of essential oils against planktonic and biofilm-forming cells of Pseudomonas aeruginosa” reports very limited and preliminary data, beyond many substantial mistakes. It’s necessary to improve the research plan, and go deeper on your main propose to compere antimicrobial activity between volatile and liquid fractions of EO. Perhaps add extra experiments and explore the inhibitory effect between EO fractions must be considered.

You should be clear what you exactly did in the manuscript through the title. You mention “The comparison…” when you only showed a brief description about anti-pseudomonal activity of volatile and liquid fractions of EO oils, any comparative analysis was done. The title should be rethink. Perhaps, “In vitro anti-Pseudomonas aeruginosa activity of volatile and liquid fractions of essential oils against planktonic and biofilm cells”. The Introduction was clear and comprehensive, covering all hot points of the proposal. 

I had problem to see the figure 1, since they didn’t appear in document. Maybe some problem has occurred during manuscript submission. I also need to highlight the way that your anti-biofilm analysis in the item 2.3, “Antimicrobial activity…P. aeruginosa biofilms”. The presented results in the figure 2 has serious flaws in the analysis. Since your assay was conducted by resazurin staining followed through absorbance you shouldn't  present those data like “cell reduction” mainly in biofilm form. The standard microbial assay able to say something about antimicrobial activity in those cases like yours is through colony count method presented as CFU/mL. Additionally, it should be considered the high amounts of exopolimeric substances (EPS) covering biofilm-encased cells interferes with the metabolic activity determination by tetrazolium salts. This is can be clearly noted by standard deviations, which many times is higher than results. In the same figure, any statistical analysis was noticed, and lack positive control to comparative analysis. It should be also avoid shown results on percentage, the impoverishes the data. 

Despite the great relevance, the manuscript presents just preliminary outcomes. Perhaps, as mentioned before, additional experiments are needed to go further in the article subject. I hope that these revisions can improve your manuscript for future submitions. 

Kind Regards,

Reviewer 2 Report

The manuscript is well organized. However, some crucial information is missing. My comments and remarks are presented in the attached file.

Reviewer 3 Report

Dear authors,

The manuscript deals with a very important topic. In general, the text is well written and the results are significant. However, there are some points that need to be improved.

1. Line 2: I don't think the comparative part of the study carries enough weight to be highlighted in the title of the manuscript.

2. Line 3: A fraction is a part of the whole. However, what was used in the tests was the integral essential oil and a volatile fraction of this oil. Therefore, instead of a "liquid fraction" what was used in the tests was the essential oil in its liquid form.

3. Line 55: "Phenols" is relative to the chemical characteristic of the compound, not its metabolite class, like terpenes. A compound can be a terpene and a phenol at the same time. The most coherent would be to write: terpenes, sesquiterpene, and phenylpropanoids.

4. Line 107: Figure 1 is missing.

5. Line 133: What criteria was used? What are the upper and lower limits for a sample to fit into one or the other category?

6. Line 216: The table does not just represent reduction, as there are cases where there was growth. It would be correct to use the term "variation in average cell size"

7. Line 254: I suggest authors start the sentence with "In addition, it is ".

8. Line 260 - 266: This part of the text needs to be further explained. Authors should explain why solubility is important in the diffusion test and then conclude on the possible impact of the difference between the solubilities of linalool and cineole.

9. Line 302: Volatility not only depends on molecular weight, but also on intermolecular interactions.
I recommend that authors cite the vapor pressures of substances, rather than molecular weights, to support their discussion.

10. Line 311: So the emulsified oil should be more active than its non-emulsified form, as the micelle would ensure the oil droplets disperse in a hydrophilic medium (the cell wall) better than the free oil.

11. Materials and Methods: Any comparison between results requires statistical analysis. In the work, the authors discuss the greater or lesser activity of the samples without presenting statistical calculations. That is inadmissible.

12. Line 564 and 568: Please correct the numbering of items 4.72 and 4.10.1.

Round 2

Reviewer 2 Report

I would suggest introducing to the main body:

11.      Information on no significant changes in emulsion parameters as a result of the dilution for MIC and MBEC assays (sections 4.6.2 and 4.6.3, respectively) with reference presented in the cover letter.

22.      Reference to the methodology of Inverted Patri Dish (added information in lines 620-622). The reference is presented in the cover letter but was not introduced into the text.

Reviewer 3 Report

Dear authors,

Thank you for accepting my suggestions. In my opinion the manuscript is now ready to be accepted for publication in Molecules.